# Meta-Analysis of the Effect of Different Exercise Mode on Carotid Atherosclerosis

**DOI:** 10.3390/ijerph20032189

**Published:** 2023-01-25

**Authors:** Pincao Gao, Xinxin Zhang, Shanshan Yin, Haowen Tuo, Qihan Lin, Fang Tang, Weiguo Liu

**Affiliations:** 1College of Physical Education and Health, Guangxi Normal University, Guilin 541006, China; 2College of Rehabilitation and Health, Hunan University of Medicine, Huaihua 418000, China; 3Minstry of Public Sport, TaiZhou University, Taizhou 225300, China

**Keywords:** exercise, carotid atherosclerosis, carotid intima-media thickness, total cholesterol, low-density lipoprotein, high-density lipoprotein

## Abstract

(1) Background: There is increasing evidence showing the health benefits of exercise on carotid atherosclerosis. However, little is known about the different exercise modes for carotid atherosclerosis. This study was designed to perform a meta-analysis of effect of different exercise modes on carotid atherosclerosis so as to provide evidence-based suggestions for the prevention and management of cardiovascular and cerebrovascular diseases. (2) Methods: Six databases were systematically searched to identify randomized trials that compared exercise to a non-exercise intervention in patient with carotid atherosclerosis. We a priori specified changes in cIMT, TC, LDL-C, and HDL-C biomarkers as outcomes. (3) Results: Thirty-four trials met the eligibility criteria, comprising 2420 participants. The main analyses showed pronounced differences on cIMT (MD = −0.06, 95%CI (−0.09, −0.04), *p* < 0.00001, TC (MD = −0.41, 95%CI (−0.58, −0.23), *p* < 0.00001), LDL-C (MD = −0.31, 95%CI (−0.43, −0.20), *p* < 0.00001), and HDL-C (MD = 0.11, 95%CI (0.04, 0.19), *p* = 0.004), which significantly reduced the risk factors of carotid atherosclerosis disease. In the different exercise modes, the effect was pronounced for aerobic exercise for all outcomes except TC; high-intensity interval exercise also showed significance for all outcomes except TC and HDL-C; aerobic exercise combined with resistance exercise did not affect any outcome except HDL-C; (4) Conclusions: Exercise has a prominent prevention and improvement effect on carotid atherosclerosis. In the perspective of exercise pattern, aerobic exercise and high-intensity intermittent exercise can improve carotid atherosclerosis; however, aerobic exercise has a more comprehensive improvement effect.

## 1. Introduction

Arteriosclerosis is a non-inflammatory disease of arteries, which can lead to serious cardiovascular diseases, cerebrovascular diseases, peripheral arterial disease, and type 2 diabetes and greatly threaten human life and health. Most arteriosclerosis is located in the major and medium arteries, including the coronary artery, carotid artery, and brain base ring. In 2020, the global prevalence of carotid plaque in the population aged 30–79 was estimated at 21.1% [1], and about 8.5 million people over 40 years old have peripheral arterial disease in USA every year [2] due to arteriosclerosis. Moreover, ischemic stroke caused by carotid arteriosclerosis accounts for 15% to 20% of cerebrovascular disease, and the more severe the carotid artery stenosis, the higher the risk of stroke and the more severe the disease [3].

Atherosclerosis is the pathological basis of cardiovascular disease, while the formation of atherosclerosis is related to dyslipidemia [4,5]. However, regarding how to prevent and treat atherosclerosis, the state of the reduction of its incidence rate is still in the exploratory stage. Currently, carotid endarterectomy, carotid stent implantation, and intensive drug therapy are the main treatment methods for carotid atherosclerosis. With in-depth study of the disease mechanism, it has been found that exercise combined with drug therapy has better effects on the prevention of atherosclerotic disease, the stabilization, and even the reversal of plaque [6]. In fact, mounting evidence suggests that exercise plays a positive role in improving atherosclerosis [7,8]. Schroeder et al. showed that 8 weeks of combined training might provide more comprehensive CVD benefits compared to time-matched aerobic or resistance training alone [9].

Although there are many randomized controlled exercise trials on atherosclerosis, it is unclear which exercise method is better for the treatment of carotid atherosclerosis due to different exercise methods, different outcome evaluations, different research objectives, and inconsistent results. Therefore, this study used a meta-analysis to study the intervention effect of different exercise methods on carotid atherosclerosis to provide evidence for the formulation of exercise prescriptions for atherosclerotic diseases.

## 2. Materials and Methods

### 2.1. Protocol and Registration

This review followed the preferred Reporting Items for Systematic Reviews and Meta-analyses (PRISMA) guidelines 9, and it also was registered on the *PROSPERO* database (Systematic Review Registration: https://www.crd.york.ac.uk/prospero/#myprospero:CRD42021260832) (accessed on 14 July 2021).

### 2.2. Ethics

Approval from a human ethics committee was not required for this research since this study was a systematic review.

### 2.3. Search Strategy

*PubMed*, *Embase*, *Medline*, *Cochrane*, *CNKI*, and the Chinese *WanFang* databases were searched from the earliest date until December 2022 using searches with medical subject headings (mesh) term combinations related to exercise and atherosclerosis. The keywords were exercise (or exercise intervention) and atherosclerosis (or atherosclerotic vascular disease), exercise (or exercise intervention), and carotid atherosclerosis (or carotid atherosclerosis plaques). In addition, the search strategy of this study uses a combination of mesh terms and keywords. It was determined after repeated checks, supplemented by manual search, and retrospectively included references when necessary.

### 2.4. Inclusion and Exclusion Criteria

#### 2.4.1. Inclusion Criteria

Included studies were randomized controlled trials that compared an exercise intervention to a non-exercise control group in patients with carotid atherosclerotic. Articles written in English and Chinese languages were included. The baseline interventions included routine medication treatment and diet control and were equally implemented in both groups. Otherwise, the exercise intervention in the experimental group included aerobic exercise, resistance exercise, and high-intensity interval training or a combination of these modes, while the control group had no exercise intervention.

#### 2.4.2. Exclusion Criteria

Animal experiments, case reports, conference abstract reviews, and qualitative studies were excluded. Documents with incomplete data or data problems or inconsistent main outcome indicators were excluded.

### 2.5. Outcomes

The outcomes included carotid intima-media thickness (cIMT), total cholesterol (TC), low-density lipoprotein cholesterol (LDL-C), and high-density lipoprotein cholesterol (HDL-C).

### 2.6. Data Extraction and Synthesis

Two reviewers (P.G. and X.Z.) performed data extraction independently in prespecified forms and then cross-checked. The extracted contents included the basic materials about author, the year of publication, number of participants, intervention design, type, frequency, duration, clinical outcome measures, etc. In addition, specific details of experimental design were also performed by three reviewers (H.T., Q.L., and S.Y.), such as randomization, allocation and hiding, blind method, basic data, intervention measures, outcomes, intervention time, and follow-up time of the study subjects. Supposing that RCTs of multiple studies were involved, the experimental and control groups related to this study were extracted. Eventually, the changes between post intervention and pre intervention were extracted as the baseline value of continuous outcomes (mean ± standard deviation, SD) into an electronic database. Any conflicts in the reported methods or results occurring during the data extraction process were arbitrated by a fourth reviewer (F.T.) and resolved by consensus.

### 2.7. Literature Quality Evaluation

We used the RoB2 to conduct an overall assessment of risk of bias in this systematic review [10], and “low risk bias”, “high risk bias”, and “unclear” (lack of relevant information or uncertainty of bias) were assessed for all included literatures. The quality evaluation of the literature was conducted independently by two reviewers (W.L. and F.T.). Otherwise, Jadad score was used to assess study quality [11], with a total score of 7 points. Scores ≥ 4 were considered high-quality studies, while scores < 4 were considered low-quality studies.

### 2.8. Statistical Analyses

Data analyses were conducted by the RevMan5.3 software (Cochrane, London, United Kingdom). The experimental data were continuous variables, and mean difference (MD) and 95%confidence intervals (95%CI) were used as effect scales to combine effect sizes. Heterogeneity test was performed using the Q statistic, and if *p* < 0.05, and I^2^ ≤ 50%, indicating that the studies are homogeneous, then a fixed-effect model was used for analysis. If *p* ≤ 0.05, and I^2^ > 50%, indicating statistically heterogeneity, then a random-effect model was used for analysis. Finally, the Egger’s regression asymmetry test was used to detect publication bias.

## 3. Results

### 3.1. Search Results

Figure 1 shows the flow diagram of the selection process of the literature. In total, 3405 potentially eligible articles were obtained by searching various databases and then were checked manually. A total of 1936 documents remained by removing duplicate records from EndNote x9 software. Next, 1686 irrelevant studies were excluded by preliminary screening of the titles and abstracts of the literature, leaving 223 remaining documents. We excluded 189 of these based on further reading of the full text, with the common reasons for exclusion including having non-RCT design, unrelated outcomes, or no continuous exercise intervention or no control group, etc. Finally, a total of 34 RCT articles were deemed eligible for inclusion and quantitative synthesis for the meta-analysis.

### 3.2. Study Characteristics

The basic characteristics of the included studies are shown in Table 1. All included articles were published between 2004 and 2020. A total of 2420 individuals were involved in the 34 eligible articles were that were included in this meta-analysis. In this meta-analysis, 21 studies were published in English, and 13 studies were published in Chinese. Trial sample sizes ranged from 21 to 160 participants. Regarding exercise intervention mode, 25 involved aerobic exercise intervention, 8 involved a high-intensity interval exercise, and 6 involved aerobic exercise combined with resistance exercise. All the control groups of the included articles did not receive exercise intervention. There were 28 studies deemed to have high quality and 6 studies low quality after assessing the quality of each study by the Jadad scale.

### 3.3. Risk of Bias

The risk of bias of all included articles was evaluated by the Cochrane Collaboration’s Risk of Bias 2 (RoB2) tool, and these results are summarized in Figure 2 and Figure 3. Overall, 31 of 34 (91.2%) trials described the process of random sequence generation, and they were low-risk in the fields of random sequence generation. Most studies were classified as having an unclear risk in allocation concealment, and only three were low-risk. A high risk of bias was detected in the domain of blinding of participants and personnel; only six were at low risk; a low risk of bias was observed in blinded outcome assessment except five of them that were at high risk of bias. A low risk of incomplete outcome data bias was observed in all of the studies (34, 100%). With regard to selective outcome reporting bias, most studies were determined as low-risk. All studies were graded as unclear risk of other bias. These results are summarized in Figure 2.

### 3.4. Effects of Different Exercise Mode on cIMT

Carotid intima-media thickness (cIMT) is a crucial risk factor for cardiovascular health. A total of 26 articles used cIMT to evaluate the therapeutic effect of exercise on carotid atherosclerosis [12,13,14,15,17,18,19,20,22,23,26,27,28,29,30,31,33,34,35,37,38,39,40,41,43,44]. The random-effects model was performed to integrate the results. The results showed that, overall, exercise significantly reduced cIMT (MD = −0.06, 95%CI (−0.09, −0.04), *p* < 0.00001; Figure 4), which was statistically significant compared with the control group. The I-squared (I^2^) of this results as > 60%, indicating high heterogeneity; therefore, we performed a subgroup analysis based on exercise patterns to discuss the source of heterogeneity. In order to further explore the source of heterogeneity, we further classified and excluded the original literature, and the results showed that I2 of the cIMT effect size could be reduced to 58% if Adams 2017, Kadoglou 2013 (E2), Wang 2018 and Zhang 2012, and Zhang 2020 were excluded. However, we found that there were no differences in the sample size and the intervention of these articles compared to the other literature.

Thirty studies were conducted as a subgroup analysis of exercise intervention patterns, including 20 on aerobic exercise [14,15,20,22,23,26,27,28,29,30,31,33,36,37,38,39,40,41,43,44] 6 on high-intensity interval exercise [12,18,20,23,27,34] and 6 on aerobic exercise combined with resistance exercise [13,17,19,26,30,35]. The results showed that cIMT was decreased significantly by aerobic exercise intervention (MD = −0.08, 95%CI (−0.12, −0.05), *p* < 0.00001; Figure 5) and high-intensity interval training (MD = −0.06, 95%CI (−0.09, −0.02), *p* = 0.001; Figure 5), with statistical significance. However, cIMT was not decreased by aerobic exercise combined with resistance exercise (MD = −0.02, 95%CI (−0.05, −0.01), *p* = 0.16; Figure 5), but the changes were not statistically significant.

The heterogeneity of subgroup analyses was also prominent from the research results. To explore the source of heterogeneity of the subgroup, sensitivity scores were used to analyze the excluded studies one by one and evaluate the cIMT effect size of each study. However, the results showed that there was little difference in heterogeneity among different studies, and the elimination of any article had little influence on the effect size of cIMT, and the results of the meta-analysis were relatively stable.

### 3.5. Effects of Different Exercise Mode on TC

Twenty-two trials used TC to evaluate the clinical effect of exercise on atherosclerosi s [12,13,16,17,18,19,20,21,23,24,25,26,27,28,32,34,36,38,39,40,42,43]. A fixed- effect model was used for merge the results. The overall effect of the studies showed that exercise significantly decreased TC compared with the control group (MD = −0.41, 95%CI (−0.58, −0.23), *p* < 0.00001; Figure 6). The I^2^ of analysis was 92%, indicating high heterogeneity. We found that the result of I^2^ could be reduced to 57% if Hasegawa 2018 (E1), Hasegawa 2018 (E2), Kim 2017 (E1), and Zhang 2012 were excluded. However, no differences were found in the sample size, the intervention time, drug usage, and health condition of these studies compared to the other literature.

In the different exercise modes of the subgroup of meta-analysis, there was no effect on TC by intervention of aerobic exercise (MD = −0.27, 95%CI (−0.55, −0.02), *p* = 0.07; Figure 7) [16,17,20,21,25,26,27,28,32,36,38,39,40,42,43,45], high-intensity interval training (MD = −0.42, 95%CI (−0.89, −0.04), *p* = 0.07; Figure 7) [12,18,20,23,24,25] and aerobic exercise combined with resistance exercise (MD = 0.22, 95%CI (−0.67, 0.24), *p* = 0.36; Figure 7) [13,17,19,26,35].

### 3.6. Effects of Different Exercise Mode on LDL-C

Twenty-four studies used the LDL-C to evaluate patients with atherosclerosis [12,13,16,17,18,19,20,21,22,23,24,26,27,28,29,31,32,34,35,36,39,42,43,45]. The random-effects analysis was managed to merge the results (I^2^ =84%). The results showed that exercise markedly decreased the LDL-C compared with the control group (MD = −0.31, 95%CI (−0.43, −0.20), *p* < 0.00001; Figure 8). The result of I^2^ was 84%, and if Byrkjeland 2016, Fan 2008 (E1), Fan 2008 (E2), Li2018, and Liu 2021 were excluded, the results of I^2^ could drop down to 50%. The interventions of aerobic exercise plus taking medicine or aerobic exercise plus diet or a combination of aerobic exercise and resistance exercise in these studies were found by further analysis, and these interventions may be the cause of heterogeneity.

Subgroups of meta-analysis were conducted for exercise intervention, and the results showed that the effects on LDL-C were significantly decreased by intervention of aerobic exercise (MD = −0.34, 95%CI (−0.50, −0.18), *p* < 0.0001; Figure 9) [16,17,21,22,24,25,27,28,29,31,32,34,36,39,42,43,45] and high-intensity interval training [MD= −0.47, 95%CI (−0.74, −0.21) *p* < 0.0001; Figure 9) [12,18,20,23,24,34]; nevertheless, no significant differences of LDL-C were observed in aerobic exercise combined with resistance exercise (MD = −0.12, 95%CI (−0.46, 0.22), *p* = 0.49; Figure 9) [13,17,19,26,35]. Further, little influence on LDL-C effect size was shown by eliminating a certain article in sensitivity analysis, so the results of the meta-analysis were relatively stable.

### 3.7. Effects of Different Exercise Mode on HDL-C

Thirty studies used HDL-C to evaluate patients with atherosclerosis, including 17 aerobic exercise, 7 high-intensity interval training, and 5 aerobic exercise combined with resistance exercise [12,13,16,17,18,19,21,22,23,24,25,26,27,28,29,31,32,34,35,36,39,42,43]. The fixed-effects analysis was conducted to merge the results (I^2^ = 87%). The results showed that exercise markedly improved the HDL-C compared with the control group (MD = 0.11, 95%CI (0.04, 0.19), *p* = 0.004; Figure 10). If studies including Adams 2017, Choi 2012, Kim 2017 (E1), Kim 2017 (E2), Li 2018, and Wang 2018 were excluded, the results of I^2^ could fall to 36%. No differences were found in the sample size, the intervention time, drug usage, and health condition of these studies compared to the other literature.

Subgroups of meta-analysis were conducted for exercise intervention, and the results showed that the effects on HDL-C were significantly increased by intervention of aerobic exercise (MD = 0.16, 95%CI (0.05, 0.27), *p* = 0.006; Figure 11) [16,17,18,19,21,22,25,26,27,28,29,31,32,36,39,42,43], and aerobic exercise combined with resistance exercise (MD = 0.23, 95%CI (0.08, 0.38), *p* = 0.003; Figure 11) [13,17,19,26,35]; nevertheless, no significant differences of HDL-C were observed in high-intensity interval training (MD = −0.00, 95%CI (−0.22, 0.22), *p* = 0.98; Figure 11) [12,18,23,24,25,27,34].

Although there was high heterogeneity of the effect of exercise on HDL-C, we found that it also has little influence on HDL-C effect size by eliminating any certain article in sensitivity analysis, so the analysis results of the meta-analysis were relatively stable.

### 3.8. Adverse Events, Sensitivity Analysis, and Publication Bias

Adverse events were not found in all included studies. Hence, this information could not be searched from the RCTs analyzed. The results of this study had high heterogeneity. By removing single studies, the sensitivity analyses showed no obvious changes in the statistical significance of all primary or secondary outcomes. The publication bias of the results in the meta-analysis was evaluated using the Egger’s test (Figure 12, Figure 13, Figure 14 and Figure 15). The Egger’s test of cIMT and HDL-C was 0.987 and 0.702, indicating there is no publication bias; however, the Egger’s test results of TC and LDL-C are 0.009 and 0.014, which are lower than 0.05, suggesting that there were a certain publication bias.

## 4. Discussion

This study was designed to evaluate the effects of aerobic exercise, resistance exercise, high-intensity interval training and combined exercise on atherosclerosis, focusing on risk reduction for individuals who had not yet progressed to cardiovascular and cerebrovascular diseases.

cIMT was the crucial indicator for evaluating carotid atherosclerosis. Studies have shown that cIMT thickening was the early clinical manifestation of atherosclerosis [46]. Mounting evidence indicates that exercise produces significant physiological and health benefits and prevents or delays the development of atherosclerosis in humans. From the perspective of evidence-based medicine, this meta-analysis showed that exercise can obviously reduce cIMTs, significantly reducing the risk factors of cerebrovascular disease. Seals DR and Che L’s article also confirmed that exercise training is an effective non-pharmacological treatment for improving carotid artery stiffness in young and older individuals [47,48]. In addition, the result of the subgroup analysis based on different exercise modes showed that cIMTs were decreased significantly after aerobic exercise and high-intensity intermittent training. No significant effect was observed on cIMT by aerobic exercise combined with resistance exercise. Carpio-Rivera E et al. showed that regular physical activity has potential benefits for arterial elasticity, especially aerobic exercise [49]. Evidence from a recent meta-analysis also suggests that aerobic training is the most effective type of exercise modality to improve blood pressure and arterial stiffness [50]. These were consistent with our research results. Our study also found that high-intensity interval training can improve carotid atherosclerosis as well as aerobic exercise.

The mechanisms by which physical activity counteracts arterial stiffening are not well-known. In order to find the intervention mechanism of aerobic exercise on carotid atherosclerosis, we further explored the effect of exercise on lipidemia metabolism. Because dyslipidemia was considered as a critical risk factor for atherosclerosis, including TC, LDL-C, and HDL-C, the formation of atherosclerosis was related to the deposition of a large amount of TC in blood vessels [51,52]. Reducing the deposition of TC in blood vessels can lessen the formation of atherosclerosis. LDL-C protein particles can carry cholesterol; if LDL-C is excessive, the cholesterol carried by LDL-C will accumulate on the arterial wall, leading to atherosclerosis [26]. In addition, the occurrence of atherosclerosis was negatively correlated with the serum HDL-C in the human body, so this is an important way to inhibit the formation of atherosclerosis by improving HDL-C level. Both clinical drug therapy or exercise intervention mainly focus on regulating blood lipids in the treatment of atherosclerosis [53]. Many studies have shown that exercise intervention can greatly improve blood lipids and lipid metabolism levels, thus further improving the formation of atherosclerosis [54].

The results of this meta-analysis showed that exercise intervention could significantly decrease the content of TC and LDL-C and increase the level of HDL-C to prevent and improve atherosclerosis [55,56]. The results of the subgroup meta-analysis showed that aerobic exercise had the remarkable effect of reducing LDL-C and increasing HDL-C. High-intensity intermittent exercise has a better effect in reducing LDL-C, but it has no obvious effect on TC and HDL-C. Aerobic exercise combined with resistance exercise showed a significant effect on HDL-C but not TC and LDL-C. The Egger’s analysis of this study shows that there is publication bias in the improvement effect of exercise intervention on TC and LDL-C, and there is no significant publication bias in the improvement effect of cIMT and HDL-C, and the meta-analysis results were relatively stable. These findings further confirm that aerobic exercise can prevent the formation of atherosclerosis by improving dyslipidemia [12,15,41], and high-intensity intermittent exercise may also play a certain role in regulating dyslipidemia.

According to the above research results, the physiological mechanism of aerobic exercise intervention on carotid atherosclerosis includes the following two aspects: first of all, exercise changes the habits of sedentary and reduces the level of risk factors causing the disease of atherosclerosis [8]; secondly, aerobic exercise can accelerate and improve the activity of the lipoprotein enzyme in the body and the metabolic decomposition of TC and LDL-C, reduce the total blood lipids, and increase the level of high-density lipoprotein [57,58] so as to further prevent and improve atherosclerosis.

### 4.1. Limitations

The study also has some notable limitations. Among the RCTs included, there was great heterogeneity with respect to exercise intervention modes, medicine intake, lifestyle, and low-quality data that may have contributed to unwanted heterogeneity. Moreover, studies with aerobic exercise combined with resistance exercise accounted for less than 18% of included articles, and there were six RCTs with combined exercise data; therefore, medical evidence on the intervention effect of high-intensity intermittent training and combined exercise on atherosclerosis needs to be further explored.

### 4.2. Practical Implications

The results of this study indicate that the intervention effect of aerobic exercise on carotid atherosclerosis is relatively stable, which can be used to guide patients to improve their condition, reduce the risk of cardiovascular disease, and thus improve their quality of life. In order to prevent and improve carotid atherosclerosis more effectively, the prescription of aerobic exercise for atherosclerosis was induced by tracing the original research literature. The minimum standard of aerobic exercise for atherosclerosis was an 8-week intervention period, 60 min of cumulative exercise time per week, and 50–70% HR peak exercise intensity. Secondly, high-intensity intermittent training can also be adopted for atherosclerosis with young patients. However, the intervention effect of aerobic exercise is more stable, and aerobic exercise should be the main intervention.

## 5. Conclusions

Exercise can significantly reduce cIMT, TC, and LDL-C and increase HDL-C, which has a good prevention and improvement effect on carotid atherosclerosis. From the perspective of exercise intervention patterns, aerobic exercise and high-intensity intermittent exercise can improve carotid atherosclerosis; however, aerobic exercise has a more comprehensive improvement effect.

## Figures and Tables

**Figure 1 ijerph-20-02189-f001:**
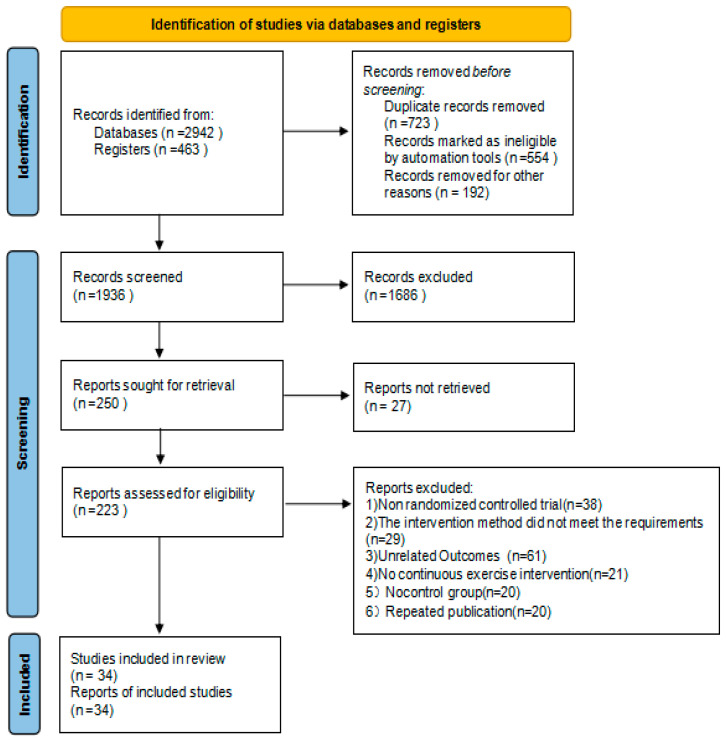
The flow diagram of the selection process.

**Figure 2 ijerph-20-02189-f002:**
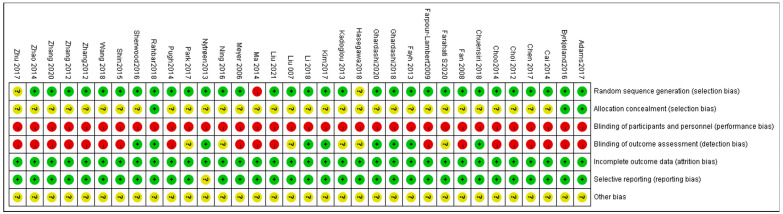
Risk of bias summary: review authors’ judgments of bias items for each included study [12,13,14,15,16,17,18,19,20,21,22,23,24,25,26,27,28,29,30,31,32,33,34,35,36,37,38,39,40,41,42,43,44,45].

**Figure 3 ijerph-20-02189-f003:**
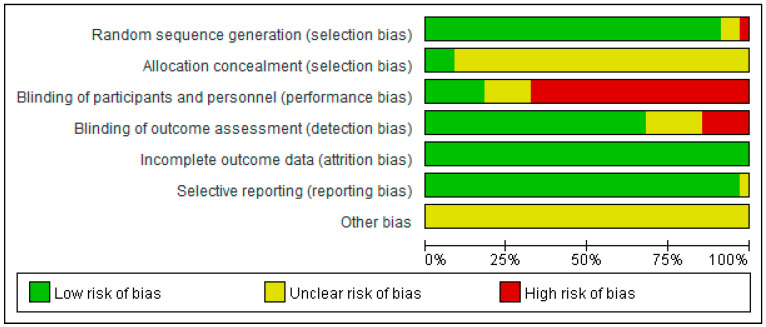
Risk of bias graph: reviewers’ judgments of each bias item, presented as percentages.

**Figure 4 ijerph-20-02189-f004:**
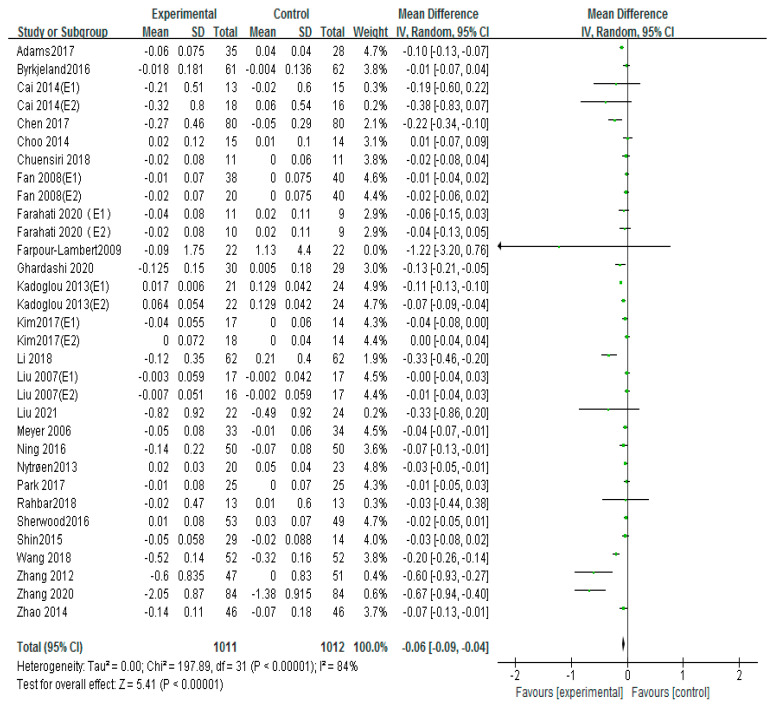
Meta−analyses of the effect of exercise on cIMT compared with the control group [12,13,14,15,17,18,19,20,22,23,26,27,28,29,30,31,33,34,35,37,38,39,40,41,43,44].

**Figure 5 ijerph-20-02189-f005:**
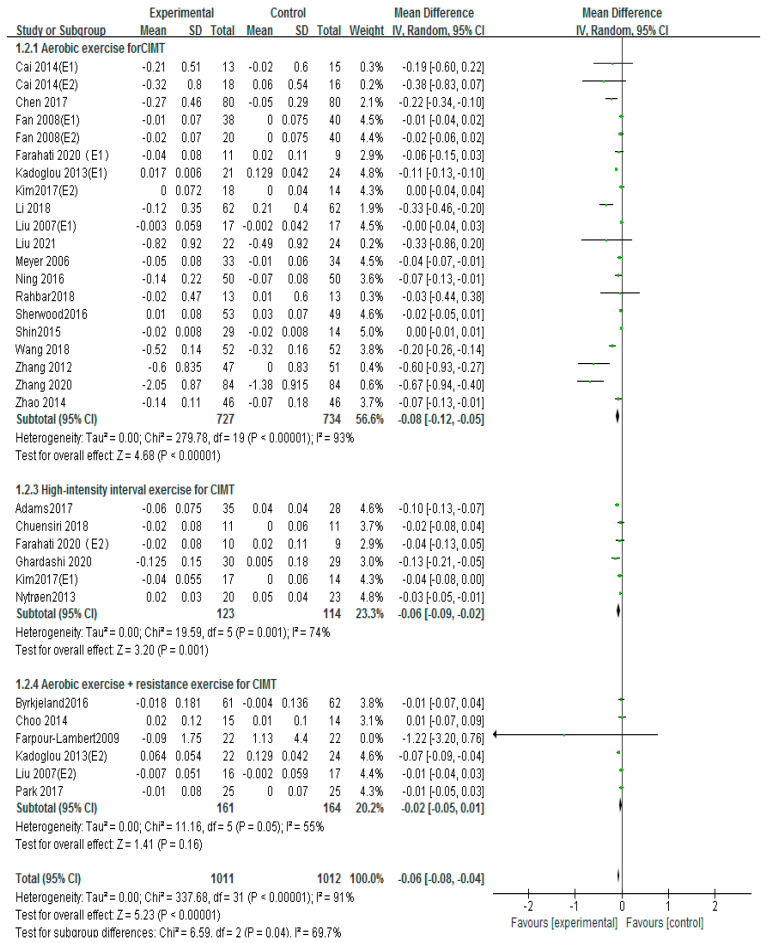
Subgroup analysis of cIMT effect size under different modes of exercise [12,13,14,15,17,18,19,20,22,23,26,27,28,29,30,31,33,34,35,37,38,39,40,41,43,44].

**Figure 6 ijerph-20-02189-f006:**
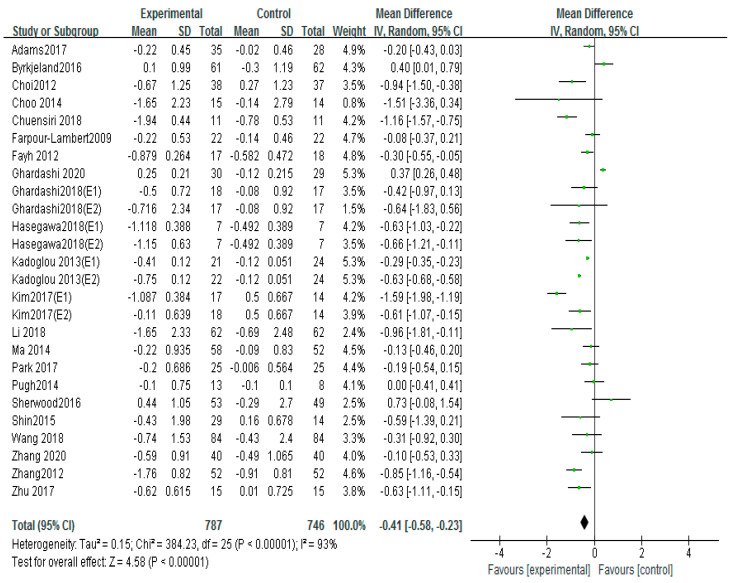
Meta−analyses of the effect of exercise on TC compared with the control groups [12,13,16,17,18,19,20,21,23,24,25,26,27,28,32,34,36,38,39,40,42,43].

**Figure 7 ijerph-20-02189-f007:**
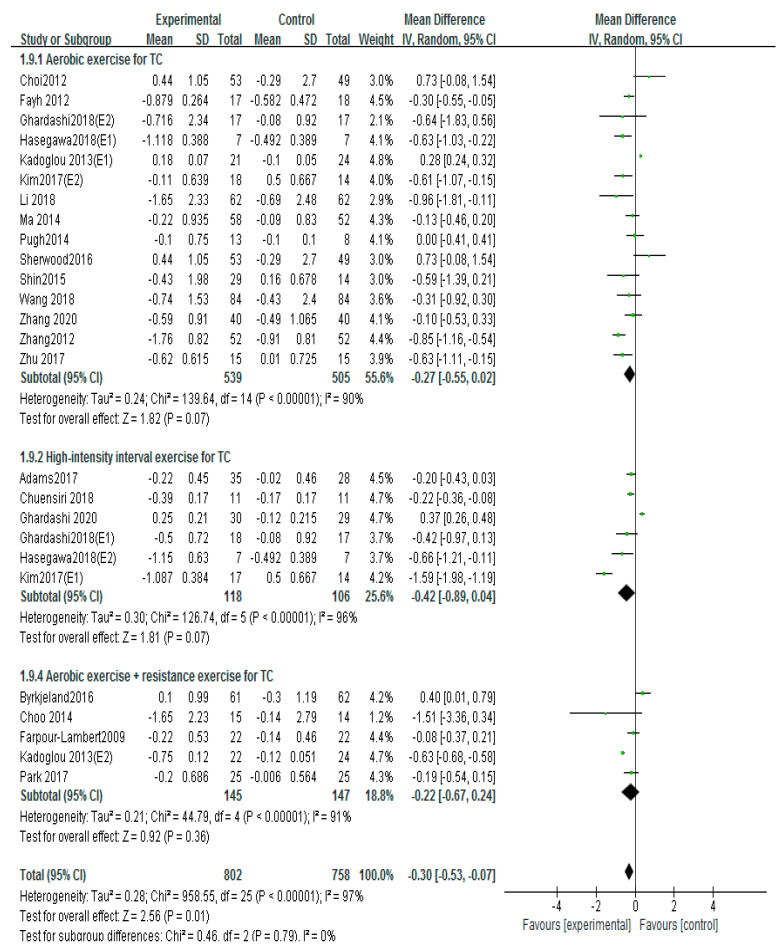
Subgroup analysis of TC effect size under different modes of exercise [12,13,16,17,18,19,20,21,23,24,25,26,27,28,32,34,36,38,39,40,42,43].

**Figure 8 ijerph-20-02189-f008:**
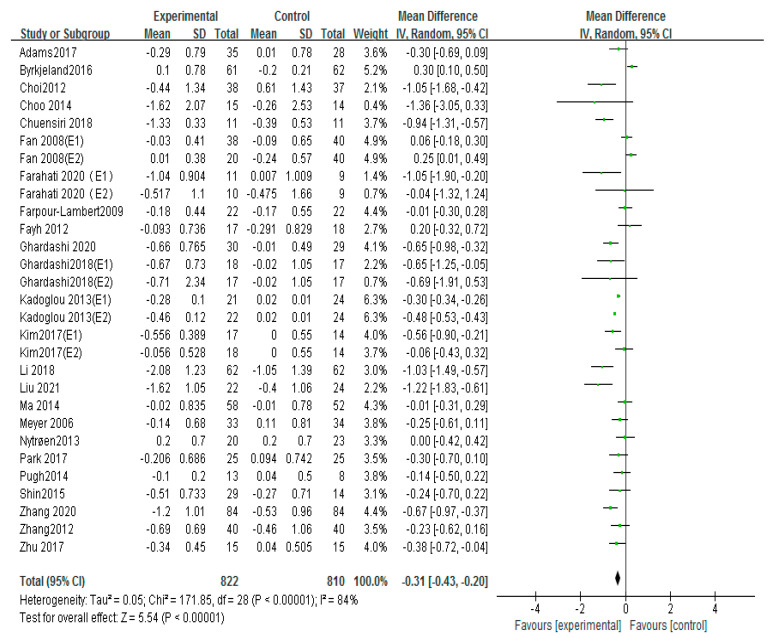
Meta−analyses of the effect of exercise on LDL−C compared with the control group [12,13,16,17,18,19,20,21,22,23,24,26,27,28,29,31,32,34,35,36,39,42,43,45].

**Figure 9 ijerph-20-02189-f009:**
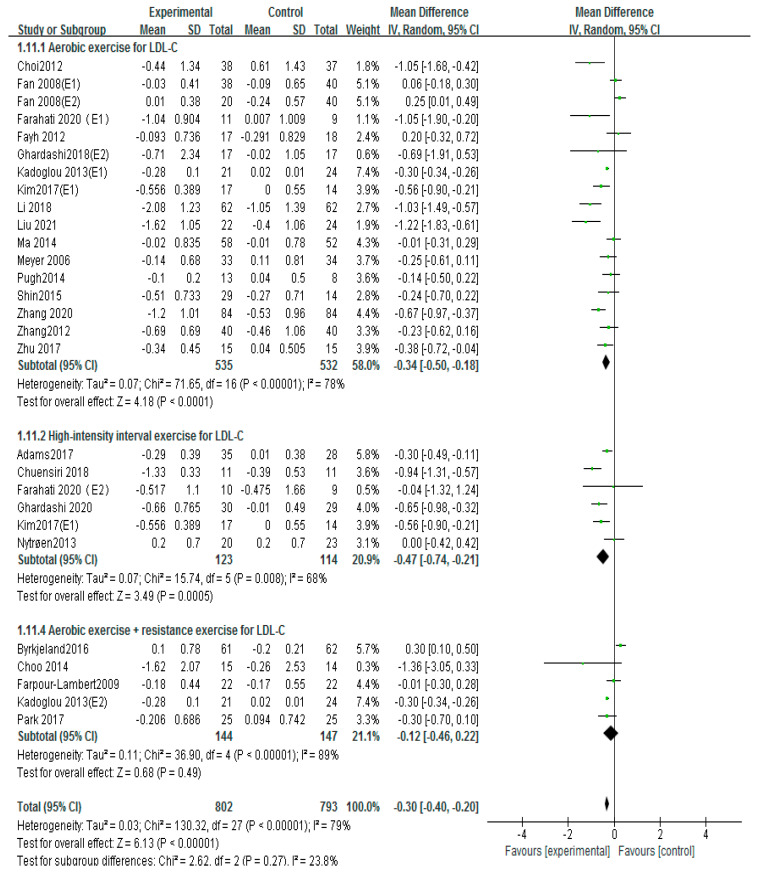
Subgroup analysis of LDL−C effect size under different modes of exercise [12,13,16,17,18,19,20,21,22,23,24,26,27,28,29,31,32,34,35,36,39,42,43,45].

**Figure 10 ijerph-20-02189-f010:**
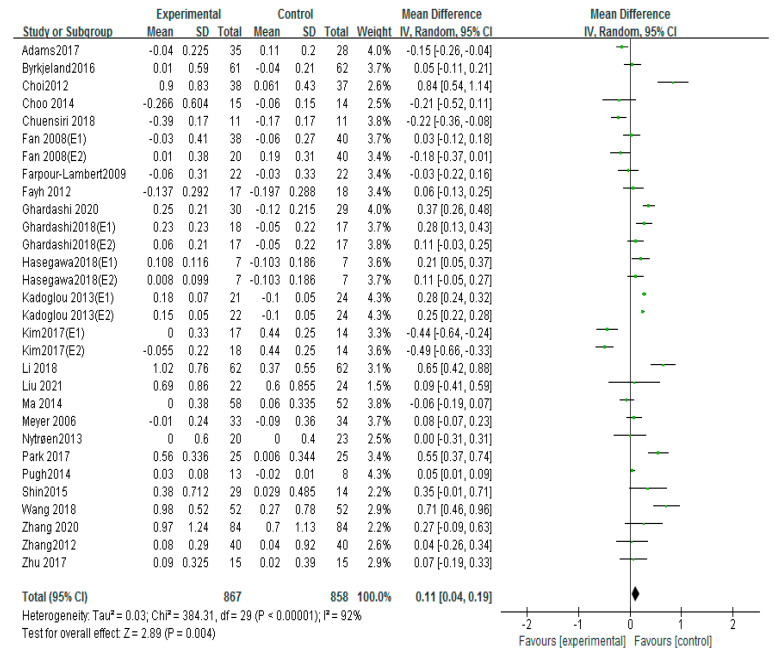
Meta−analyses of the effect of exercise on HDL−C compared with the control group [12,13,16,17,18,19,21,22,23,24,25,26,27,28,29,31,32,34,35,36,39,42,43].

**Figure 11 ijerph-20-02189-f011:**
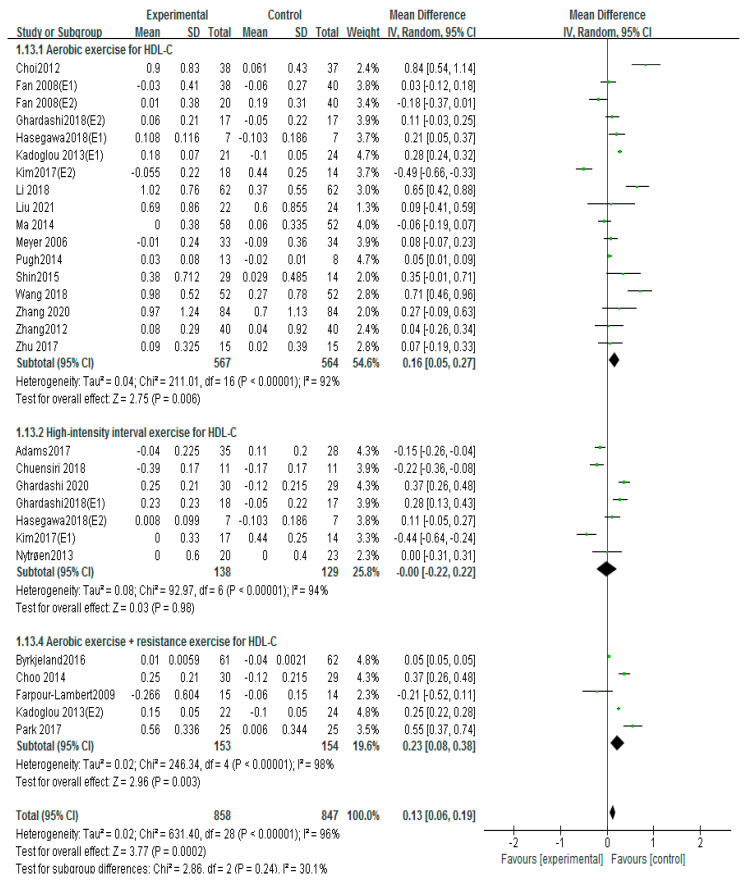
Subgroup analysis of HDL−C effect size under different modes of exercise [12,13,16,17,18,19,21,22,23,24,25,26,27,28,29,31,32,34,35,36,39,42,43].

**Figure 12 ijerph-20-02189-f012:**
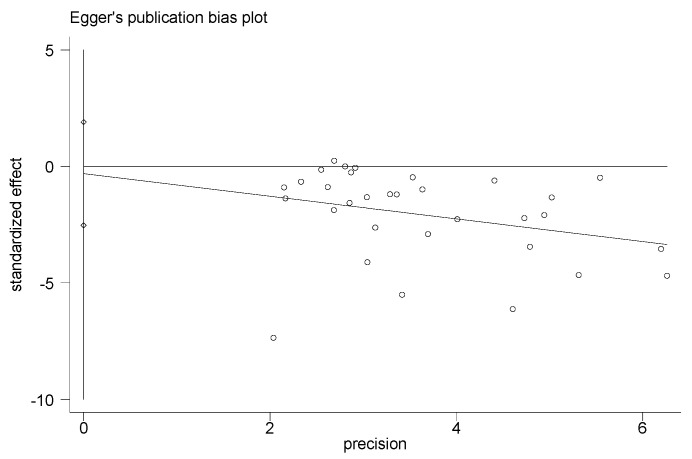
Egger’s test for evaluating the publication bias of cIMT.

**Figure 13 ijerph-20-02189-f013:**
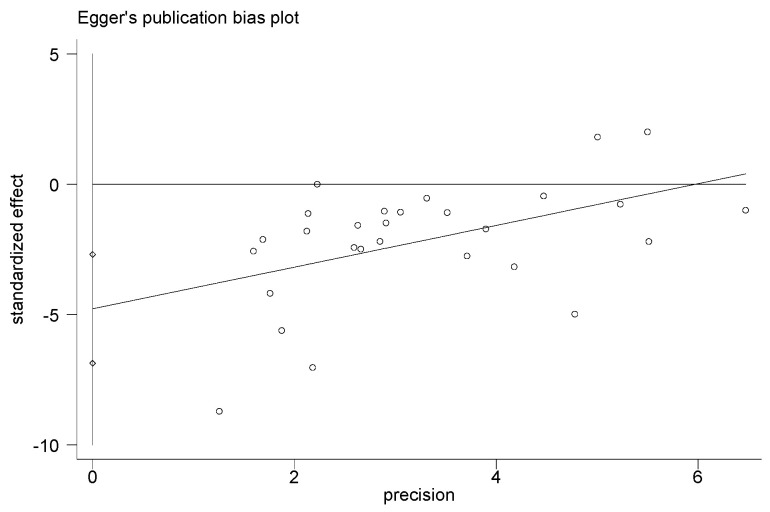
Egger’s test for evaluating the publication bias of TC.

**Figure 14 ijerph-20-02189-f014:**
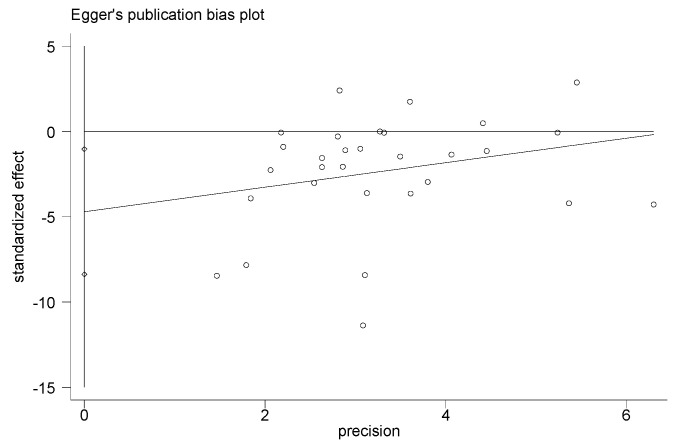
Funnel plot for evaluating the publication bias of LDL−C.

**Figure 15 ijerph-20-02189-f015:**
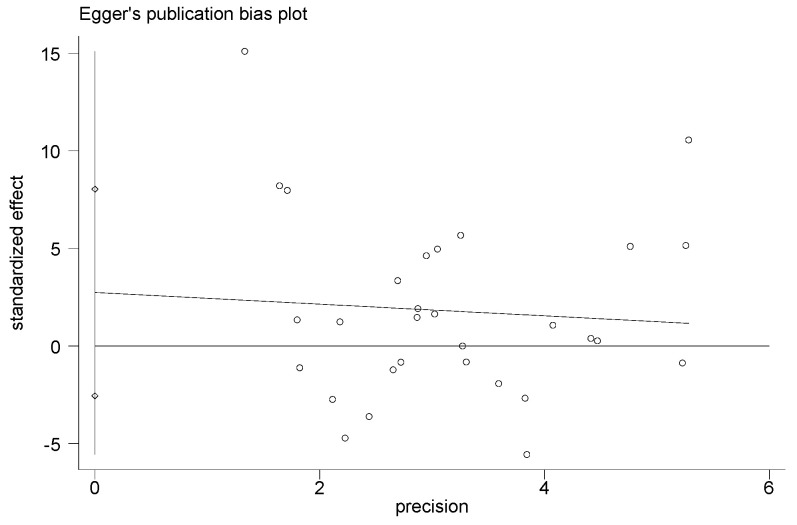
Funnel plot for evaluating the publication bias of HDL−C.

**Table 1 ijerph-20-02189-t001:** The detailed characteristics of each selected study.

Author, Year	Country Language	Subject Type	Sample (E/C)	Mean Age/Year (E/C)	Intervention Program (E/C)	Exercise Intensity	Duration of Intervention			Drug Usage	Outcome	Jadad Score
							Frequency (weekly)	Time (min)	Duration			
Adams 2017 [12]	Canada (English)	Testicular cancer survivors	E = 35 C = 28	34.3 44.0	HIIT NE	4 * 4 min 95% VO_2peak_	3	35	12 weeks	-	cIMT; TC; LDL-C; HDL-C	7
Byrkjeland 2016 [13]	Norway (English)	Patients with type 2 diabetes and coronary heart disease	E = 61 C = 62	63.5 63.2	AE + RE NE	⅔AE + ⅓RE	3	60	12 weeks	-	cIMT; TC; LDL-C; HDL-C	5
Cai 2014 [14]	China (Chinese)	Men with atherosclerotic Women with atherosclerotic	E1 = 13 C1 = 15 E2 = 18 C2 = 16	49 52 51 48	AE NE AE NE	75% HRpeak 75% HRpeak	5 5	40 40	12 weeks 12 weeks	-	cIMT	5
Chen 2017 [15]	China (Chinese)	Patients with hypertension and anxiety	E = 80 C = 80	72.35 72.26	AE NE	Less than 70% of HRmax	5	20~30	4 months	-	cIMT	3
Choi 2012 [16]	South Korea (English)	Patient with type 2 diabetes	E = 38 C = 37	53.8 55	AE NE	3.6~6.0 MET	5	60	12 months	-	TC; LDL-C; HDL-C	6
Choo 2014 [17]	South Korea (English)	Healthy women	E1 = 20 E2 = 15 C = 14	46.0 41.8 43.1	RE RE + AE NE	50–70% HRmax; Two sets of 8–12 repetitions	3	30	12 months	-	cIMT; TC; LDL-C; HDL-C	4
Chuensiri 2018 [18]	Thailand (English)	Obese preadolescent boys	E = 11 C = 11	11.0 10.6	HIIT NE	90% of peak power output	3	16	12	-	cIMT; TC; LDL-C; HDL-C	5
Farpour- Lambert 2009 [19]	Switzerland (English)	Prepubertal obese children	E = 22 C = 22	9.1 8.8	AE + RE NE	55~65% VO_2peak_, 2 to 3 groups of 10 to 15 times	3	60	12 weeks	-	cIMT; TC; LDL-C; HDL-C	6
Farahati 2020 [20]	Iran (English)	Inactive and overweight women	E1 = 10 E2 = 11 C = 9	42.8 43.9 44.2	HIIT AE NE	85–95% of HR_peak_ 60–70% of HR_max_	3	25 47	12 weeks	-	cIMT; TC; LDL-C	5
Fayh 2013 [21]	Brazil (English)	Obese	E = 17 C = 18	32.3 31.4	AE +DT NE + DT	70% HRR	3	45	65.9 days 79.7 days	-	TC; LDL-C; HDL-C	5
Fan 2008 [22]	China (Chinese)	Overweight and obese children	E1 = 18 E2 = 20 C = 40	10 10 9.9	AE +DT AE +DT NE + DT	60~70% HR_peak_ 60~70% HR_peak_	2 1~2	75 75	6 weeks 1 year	-	cIMT; LDL-C; HDL-C	3
Ghardashi2020 [23]	Iran (English)	Patients with type 2 diabetes	E = 30 C = 29	55.10 54.10	HIIT + MT NE + MT	85–90% HR_max_	3	24	12 weeks	-	cIMT; TC; LDL-C; HDL-C	5
Ghardashi 2018 [24]	Iran (English)	Patients with type 2 diabetes	E1 = 18 E2 = 17 C = 17	54.78 53.12 54.24	HIIT AE NE	12 repetitions of 1.5 min 85~95% HR_peak_ 70% HR_peak_	3 2	42 42	12 weeks 12 weeks	-	TC; LDL-C; HDL-C	6
Hasegawa 2018 [25]	Japan (English)	Healthy men	E1 = 7 E2 = 7 C = 7	23.7 23.1 20.7	HIIT AE NE	12 repetitions of 1.5 min 85~95% HRpeak 60~70% VO_2peak_	3	3.5 45	6 weeks 8 weeks	-	TC; HDL-C	5
Kadoglou 2013 [26]	Greece (English)	Patients with type 2 diabetes	E1 = 25 E2 = 25 E3 = 25 C = 25	58.3 56.1 57.9 57.9	AE RE AE + RE NE	60–75% of HRmax 60–80% load _max_	4	60	4 weeks	-	cIMT; TC; LDL-C; HDL-C	4
Kim 2017 [27]	America (English)	Sedentary elderly	E1 = 17 E2 = 18 C = 14	65 65 62	HIIT AE NE	4 * 4 min 95% HR_peak_ 70% HR_peak_	4	60	8 weeks	-	cIMT; TC; LDL-C; HDL-C	6
Li 2018 [28]	China (Chinese)	Patient with hypertensive carotid atherosclerosis	E = 62 C = 62	58.2 57.8	AE + MT NE + MT	-	3~5	35	6 months	-	cIMT; TC; LDL-C; HDL-C	4
Liu 2021 [29]	China (Chinese)	Patients with carotid atherosclerosis	E = 22 C = 24	52.96 53.00	AE + MT NE + MT	Fast walk: 3~4 km/h; Tai Chi Quan: 70% HRpeak	5	80	6 weeks	-	cIMT; LDL-C; HDL-C	5
Liu 2007 [30]	China (Chinese)	Patients with impaired glucose tolerance	E1 = 17 E2 = 12 C = 16	49.8	AE + DT AE + RE+ DT NE + DT	60~70% HRpeak 60~70% HRpeak AE + 2~3 groups, 15~20 times RE	4 4	60 50	24 weeks 24 weeks	-	cIMT	3
Meyer 2006 [31]	Germany (English)	Obese children	E = 33 C = 34	13.7 14.1	AE NE	3 sets of 8–12 reps, 80% 1 RM	3	45	6 months	-	cIMT; LDL-C; HDL-C	5
Ma 2014 [32]	China (Chinese)	Patients with type 2 diabetes	E = 58 C = 52	60.67 60.42	AE + MT NE + MT	70~80% HR_peak_	3~4	60~90	6 months	Antihypertensive drugs	TC; LDL-C; HDL-C	3
Ning 2016 [33]	China (Chinese)	Patients with mild to moderate hypertension	E = 50 C = 50	60.75 61.21	AE + MT NE + MT	6 km/h	7	30	1 year	Antihypertensive drugs	cIMT	4
Nytroen 2013 [34]	Norway (English)	Patients with heart transplant	E = 20 C = 23	51 53	HIIT + MT NE + MT	12 times of 4 min, 91.5% HR_peak_	3	-	6 months	Calcineurin inhibitor	cIMT; TC; LDL-C; HDL-C	5
Park 2017 [35]	Korea (English)	Sarcopenia obesity	E = 25 C = 25	73.5 74.7	AE + RE NE	8~15 repetitions per set 13–24 RPE	3	20~30	24 weeks	-	cIMT; TC;LDL-C; HDL-C	5
Pugh 2014 [36]	England (English)	Patients with non-alcoholic fatty liver	E = 13 C = 8	44~51 43~51	AE + MT NE + MT	30~60% HRR	3~5	30~45	16 weeks	Antihypertensive drugs	TC; LDL-C; HDL-C	5
Rahbar 2018 [37]	Iran (English)	Diabetic patients	E = 13 C = 15	48.31 48.6	AE NE	50~70% HR_peak_	3	30	8 weeks	-	cIMT	6
Sherwood 2016 [38]	America (English)	Patients with major depression	E = 51 C = 49	51.1 51.2	AE NE	70~85% HR_peak_	3	30	16 weeks	-	cIMT TC	5
Shin 2015 [39]	South Korea (English)	Patients with rheumatoid arthritis	E = 29 C = 27	64.0 62.7	AE NE	-	1	60	3 months	-	cIMT; TC; LDL-C; HDL-C	5
Wang 2018 [40]	China (Chinese)	Patients with carotid atherosclerosis	E = 52 C = 52	52.65 51.74	AE + MT NE + MT	-	3	30	12 weeks	-	cIMT;TC;	5
Zhang 2012 [41]	China (Chinese)	Patients with carotid atherosclerosis	E = 47 C = 51	35~58	AE NE	-	5~8	30~50	18 months	-	cIMT	4
Zhang 2012 [42]	China (Chinese)	Patients with type 2 diabetes	E = 40 C = 40	52.4	AE NE	80% (170-year) × HR	1~2	30–60	6 months	-	TC; LDL-C;HDL-C	5
Zhang 2020 [43]	China (Chinese)	Patients with carotid atherosclerosis	E = 84 C = 84	55.25 55.12	AE + MT NE + MT	4.5 km/h	4~5	40	12 months	-	cIMT; TC; LDL-C; HDL-C	3
Zhao 2014 [44]	China (Chinese)	Patients with mild-to-moderate hypertension	E = 46 C = 46	61.5 61.5	AE + MT NE + MT	6 km/h	7	30	1 year	Antihypertensive drugs	cIMT	4
Zhu 2017 [45]	China (Chinese)	Middle-aged and elderly women	E = 15 C = 15	59.91 60.13	AE NE	120–130 times/min	3	60	6 months	-	TC; LDL-C; HDL-C	2

Note: E, experimental group (E1, experimental group 1; E2, experimental group 2); C, control group; HIIT, high-intensity interval exercise; AE, aerobic exercise; RE, resistance exercise; NE, non-exercise; MT, medication treatment; DT, diet control; HR, heart rate; HRR, heart rate reserve; cIMT, carotid intima-media thickness; TC, total cholesterol; LDL-C, low-density lipoprotein cholesterol; HDL-C, high-density lipoprotein cholesterol.

## Data Availability

All relevant data are within the paper.

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
