# Peer review of "Meta-Analysis of the Effect of Different Exercise Mode on Carotid Atherosclerosis"

_ijerph, 2023, doi:10.3390/ijerph20032189_

Round 1

Reviewer 1 Report

The review evaluated the effect of different exercise modes on carotid atherosclerosis by changes in certain biomarkers. The review was interesting. However, some of the contents need more clarification based on the comments below.

1. The authors should mention the eight databases they used in the review, as stated in the methods section (line 76). However, in the abstract section (line 16), only six databases were used. For clarification.

2. I suggested the authors cite the reference for the cut of heterogeneity values.

3. The authors claimed that “Egger’s regression asymmetry test were used to detect publication bias” (line 131). So, the authors should cite the software used to test the Egger’s regression asymmetry test since the RevMan software did not provide the test. However, the test result was not available in the result section. Why? I recommended doing the Egger’s test to determine the small effect study of publication bias.

4. I wondered about the repeated publication articles listed in the PRISMA diagram. Were the methods or the outcome the same?

5. In the PRISMA diagram, the included study was 33, but in the result section (line 148) the included study was 34. Why?

6. The total of individuals differs between the abstract (line 20) and the results section (line 147).

7. Line 377 to 389 and line 394 to 401. These statements should be supported with evidence.

Author Response

We would like to thank all the reviewers for their deep and thorough reviews. We have seriously considered all the comments and carefully revised the manuscript in every part accordingly. All the changes in the manuscript are highlighted in red. We feel that the quality of the manuscript has been significantly improved, and we hope our revision will lead to an acceptance of our manuscript for publication in International Journal of Environmental Research and Public Health. Number-wise, answers to the comments and suggestions are as follows.

Reviewer 2 Report

Dear authors,

Your meta-analysis describes an interesting and relevant research question. The systematic review was properly conducted and the article is rather well organized.

However, there are serious methodological problems with the methodology of the meta-analyses. Exercise interventions need to be considered heterogeneous per se and this is something you mentioned in your limitations paragraph as well. Therefore, for the conduction of meta-analyses it is recommended, that at least the results of 5 studies need to be available before they can be pooled and analyzed in a meta-analytic model. (Gonnermann et al., 2015, PMID: 26040434). This methodological problem becomes obvious when you undertake the sub-group analyses and compare the results of 18 AE vs. 2 RE vs. 2 HIIT vs. 3 combined AE  RE RCTs with each other, as you did with the cIMT. This cannot be balanced. It's even worse in further subgroup meta-analysis models, e.g. in the FMD, TC, LDL-C, and HDL-C! In these meta-analysis models, only a single study is representing a full subgroup. This is not a meta-analysis! In these cases, meta-analysis is just not possible and therefore shouldn't be forced!

Moreover, the resolution of several meta-analysis models (data & forest plots) is so bad that the figures are impossible to read. Moreover, I'd strongly advise to adjust the scales of some of the forest plots to make them interpretable. The forest plots of figures 11, 13 and 15, for example, are useless in the current form.  

When looking at the major limitation that the majority of your subgroup meta-analyses is unreliable because of the small number of studies included, then I have to conclude, that the conclusion you drew from your analyses is misleading. When the result of your literature work is, that throrough subgroup analyses are impossible at this time because there are too few articles covering the effects of e.g. resistance training on carotid artherosclerosis, then this is an important and reliable finding. Conducting unreliable meta-analyses will be misleading.

Minor comments:

Introduction, page 1, lines 41 & 42: You submitted to an international journal, not a Chinese journal. Please additionally refer to worldwide numbers instead of Chinese numbers only. 

Introduction, page 1, line 43: You wrote, "Among the patients who survive after the disease". Am I right to assume that with "disease" you actually meant traumatic incident like a stroke? 

Search strategy, page 2, line 76: Please depict the search databases in detail.

Literature quality evaluation, page 3, line 117: The handbook describes the whole process of how to properly conduct a systematic review. The name of the respective risk of bias assessment tool of the Cochrane Collaboration is RoB2. 

Page 3, Search results: Please check your numbers in the flow diagram and the respective text. In the text you report 2903 potentially eligible articles while in the flow diagram it is 2905. Moreover, you report 1845 remaining articles after the first step of identification of eligible literature, while this number adds up to 1855 in the flow diagram. And in the final step you exclude 174 articles to end up with 34 included articles in the written paragraph, while these numbers in the graph are 175 and 33 respectively.

Table 1 should be horizontal format

Risk of bias, page 8, line 163: Please use "Cochrane Collaboration's Risk of Bias 2 (RoB2) tool" instead of "Cochrane Collaboration".

Risk of Bias, page 8, lines 167-168: Blinding of patients in an exercise intervention trial is impossible, so this is not a real "high RoB" but a general methodological problem in exercise intervention studies. ("I won't tell you to which group you were randomly allocated - exercise intervention or inactive control - but please start with your training now." - You see? This just doesn't work, so this is not a weakness of the included studies but a general limitation in this field of research.)

Author Response

We would like to thank all of the reviewers for their insightful and comprehensive comments. We carefully considered all of the suggestions and modified the manuscript in each section accordingly. All the changes in the manuscript are highlighted in red. After revising the manuscript according to suggestions, we feel that the context, especially the methods section, has been much more rigorous. We hope this revision will lead to an acceptance of our manuscript for publication in Journal of Environmental Research and Public Health. Here are our general and specific responses to reviewer #2's comments.

Round 2

Reviewer 2 Report

Dear authors,

Your revisions substantially improved the manuscript and I have only one final, minor comment:

On page 19, lines 346-347 you wrote, "No significant effect were observated on TC, LDL-C by aerobic exercise combined with resistance exercise." Besides the fact, that the language of this sentence should be revised, the fact that aerobic exercise combined with resistance exercise significantly increased HDL-C is missing. So please add "however, aerobic exercise combined with resistance exercise showed a significant effect on on HDL-C." (or similar) to this sentence.

Author Response

Response to Reviewer’s Comments

Thank the reviewers for the constructive suggestions.We have carefully read the ingihtful comments given by the reviewers,which are very helpful to improve the quality of our paper. All the changes in the manuscript are highlighted in red. W and we hope our revision will lead to an acceptance of our manuscript for publication in International Journal of Environmental Research and Public Health.Here are our responses to reviewer's comments.

minor comment:

On page 19, lines 346-347 you wrote, "No significant effect were observated on TC, LDL-C by aerobic exercise combined with resistance exercise." Besides the fact, that the language of this sentence should be revised, the fact that aerobic exercise combined with resistance exercise significantly increased HDL-C is missing. So please add "however, aerobic exercise combined with resistance exercise showed a significant effect on on HDL-C." (or similar) to this sentence.

Response M.C. :

Thanks ,and we have revised it in the manuscript according to your suggestion
